# Optimizing Bull Semen Cryopreservation Media Using Multivariate Statistics Approaches

**DOI:** 10.3390/ani13061077

**Published:** 2023-03-17

**Authors:** Ali Mokhtassi-Bidgoli, Mohsen Sharafi, James D. Benson

**Affiliations:** 1Department of Agronomy, Faculty of Agriculture, Tarbiat Modares University, Tehran P.O. Box 14115-336, Iran; 2Semex Alliance, Guelph, ON N1H 6J2, Canada; 3Department of Biology, University of Saskatchewan, Saskatoon, SK S7N 5E2, Canada

**Keywords:** bull diversity, pathways, post-thaw fertility, structural equation modelling, modeling, media optimization, I-optimal design

## Abstract

**Simple Summary:**

The dairy cattle industry relies heavily on artificial insemination facilitated by cryopreservation of bull semen. Cryopreservation causes a number of injuries to sperm reducing fertility and decreasing economic value. Protective extenders are media made up of a number of compounds. Most studies evaluate the impact of changing the concentration of one or two compounds at the same time. Here we use multivariate statistical methods on a large experimental dataset over twelve different compounds to identify key classes of compounds, key components of those classes, and the sensitivities of post-thaw viability on concentrations of these compounds. These statistical models and this methodology point towards improved cryopreservation protocols an improved understanding of the complex interactions among extender components.

**Abstract:**

Cryo-injury reduces post-thaw semen quality. Extender components play a protective role, but existing experimental approaches do not elucidate interactions among extender components, semen samples, and post-thaw quality. To identify optimal concentrations for 12 extender ingredients, we ran 122 experiments with an I-optimal completely random design using a large dataset from our previous study. We obtained a maximum predicted total motility of 70.56% from an I-optimal design and 73.75% from a Monte Carlo simulation. Individual bull variations were significant and interacted with extenders independently. 67% of bulls reliably preferred extender formulations to reach maximum motility. Multifactor analysis suggests that some antioxidants may offer superior protection over others. Partial least squares path modeling (PLS-PM) found the highest positive loadings for glutathione in the antioxidant class, glycerol in the CPA class, and fructose in the basic compounds class. The optimal ranges for milk, water, and ethylene glycol were extremely narrow. Egg yolk, cholesterol-loaded cyclodextrin, and nerve growth factor had medium-loading impacts. PLS-PM showed that CPA, osmoregulators, and basic components were the most efficient contributors to motility, while the antioxidant and extracellular protectant classes had less efficiency. Thus, ingredients, concentrations, and interactions of extender compounds are critical to extender formulation, especially when using multiple compounds with the same function.

## 1. Introduction

Artificial insemination using frozen-thawed sperm is a critical technology in the dairy cattle industry [1,2]. Cryopreserved semen also has a variety of applications for assisted reproductive technology as well as endangered animal preservation [3]. Cryopreservation entails different chemical and temperature steps that place sperm under many biochemical, mechanical, and ultrastructural stresses, resulting in detrimental impacts on post-thaw parameters and fertility potential [3]. The supporting media used for cryopreservation (known as extenders) play a critical role in the protection of sperm in the freeze-thaw process, and classical approaches mainly focus on improving one or two components of extenders at the same time. Understanding the effects of various compounds and their interactions within the media and between bulls requires sophisticated multiple models and statistical analysis that are rarely used [4,5]. Response surface methodology, optimization algorithms, and machine learning approaches have been applied to discover the interactions between media ingredients [4,6,7,8,9]. These methods are quick and cost-effective. For instance, Ramesh Pathy et al. [4] used the statistical tool “Plackett-Burman design” to find that egg yolk, vitamin C, and glucose were very significant for maintaining the motility of human sperm cells. Expanding beyond sperm, Lawitts and Biggers [6] applied simplex optimization for developing mouse embryo culture media and determining requirements for development, and Pi et al. [7] used four types of differential evolution (DE) algorithms to optimize the formulation of multicomponent dimethyl sulfoxide-free cryoprotectants in Jurkat cells.

I-optimal design is one of the most flexible ways to plan different types of experiments with many different continuous, categorical, and mixture variables, and it can be used with any number of runs [10]. This design is associated with better predictions, as well as improved factor effect estimate accuracy [10].

Alternatively, PLS regression has been widely used in estimating some dependent variables with different independent variables [11,12,13,14]. This analysis takes into account each individual in the population and gets around the problems of collinearity and overfitting [13,14]. Simulations based on the interactions between inputs and outputs are developed with the assistance of the Monte Carlo methodology. Morrell et al. [15] used PLS regression to identify variables related to bull fertility. Argiris et al. [11] used a structural equation model with a partial least squares method to assess the superiority of Holstein bulls as frozen semen producers. Structured equation modeling (SEM) is a good way to determine how active sets of variables interact with each other, and it may be used to assess complicated systems.

One of the main types of SEM is the PLS-PM method, which uses simple regressions to shed light on latent variables. As mentioned in Sharafi et al. [3], classical media components are used to make sperm resistant to damage during the cryopreservation process. The most common extender constituents, like cryoprotectants, antioxidants, fatty acids, sugars, and membrane stabilizers, can be thought of as latent variables or global levels. Two or more manifest variables can be related to each latent variable. Each latent variable and the corresponding manifest variables are presented as a block or table. As examples of manifest or single variables, glycerol and DMSO may be related to “cryoprotectants”; SOD and CAT may be related to “antioxidants”; fructose and glucose may be related to “sugars”; and cholesterol-loaded cyclodextrins and docosahexaenoic acid may be related to “membrane stabilizers”. Thus, integrating data from many factors may result in more accurate information and stronger inferences (classifications with lower error rates and predictions with less uncertainty) than a single property [16].

This study is a continuation of our earlier work [9], which also used some parts of these data. The objective of this study was to evaluate the single and global interactions of extender components as well as how each individual component can influence post-thaw sperm motility, both directly and indirectly. To do so, we used multiple factor analysis (MFA) and PLS path modeling (PLS-PM) as tools for evaluating post-thaw motilities at the level of a single variable and at the global level. The information obtained from this study is important for establishing the use of these new tools in future studies that aim to find the best ways to use cryopreservation protocols to keep fertility levels as high as possible.

## 2. Materials and Methods

### 2.1. Chemicals

All chemicals used for media preparation in this study were provided by Sigma (St. Louis, MO, USA) and Merck (Darmstadt, Germany) unless otherwise indicated. 

### 2.2. Animal Management and Semen Collection, Extender Preparation and Cryopreservation

As described in Tu et al. [9], semen samples were collected using an artificial vagina from 43 Holstein bulls, aged between 20 and 40 months, regularly used for breeding purposes at Semex (Quebec, QC, Canada). All animal experimental procedures were approved by the University of Saskatchewan Animal Care Committee (UAP 002CatA2018). Samples with standard quality metrics that include a concentration ≥ 1 × 10^9^ sperm/mL, motility ≥70%, and ≤15% abnormal morphology were selected for the experiments. After semen collection, samples were individually processed and divided into four equal aliquots to be diluted with four extenders each day. Semen samples were kept in a water bath at 33 °C while a preliminary analysis of fresh semen was performed to measure the concentration, motility, and morphology. Samples that did not meet minimum quality assessments (namely greater than 10^9^ sperm/mL, motility greater than 70%, and less than 15% abnormal morphology) were excluded from the study.

The standard control extender consists of tris base (hydroxymethyl-aminomethane; 2.4 g, *w*/*v*), citric acid (1 g, *w*/*v*), fructose (1 g, *w*/*v*), glycerol (7 mL *v*/*v*), (25 mg) gentamicin, 50,000 IU penicillin, and streptomycin 300 µg/mL in 100 mL distilled water. A total of 488 tris-based extenders (122 runs × 4 replicates) were prepared using different ranges of water (0–4 mL), basic tris (3–8 mL, composed of 3.02 gr tris and 1.74 gr citric acid), egg yolk (1–2 mL), milk (0–0.4 mL), fructose (0–125 mg), trehalose (0–165 mg), cholesterol loaded cyclodextrin (CLC) (2–10 mg), glutathione (0–10 uL), melatonin (5–12 uL), nerve growth factor (NGF) (0–500 ng), glycerol (0–0.75 mL), and ethylene glycol (0–0.85 mL).

To freeze semen, samples were diluted with the corresponding extenders in one step dilution protocol and then were cooled to 4 °C for a 4 h equilibration time in each medium with one-step processing, packaged in 0.25 mL French straws, and then frozen in a controlled rate freezer (Digitcool 007262, IMV, L’Aigle, France) as follows: from 4 °C to −12 °C at −4 °C/min, from −12 °C to −40 °C at −40 °C/min, and from −40 °C to −140 °C at −50 °C/min before plunging into liquid N_2_. After at least 24 h of storage, one frozen straw per individual bull was analyzed. To do that, frozen straws were thawed in a 37 °C water bath for 45 s and were immediately analyzed for total motility and progressive motility.

### 2.3. Measurement of Post-Thaw Recovery

We used total and progressive motility post-thaw motility as the main metrics in models to analyze and interpret our multiple statistics. We used a Sperm Class Analyzer (Microptic, Barcelona, Spain) that captures video at 50 frames per second. The diluted semen (2.5 µL) was placed on a pre-warmed chamber slide (33 °C, Life optic slide, 20 µM depth chamber), and motility metrics were determined using a phase-contrast microscope (Nikon, Mississauga, ON, Canada) with a 10× objective at 33 °C. The following cut-offs were applied for all CASA analyses: the range for particle size was defined as 8 to 150 µm^2^; VCL was used to characterize immotile cells (VCL < 40 µm/sec), slow cells (VCL between 40 and 80 µm/s), medium cells (VCL between 80 and 150 µm/s) and fast cells (VCL > 150µm/s); and progressive motility was defined as >85% STR. A minimum of 300 spermatozoa from 8 fields in 50 frames of video were acquired for each analysis.

### 2.4. Experimental Design and Multiple Statistics Analysis

An I-optimal, completely randomized design with a quadratic model and 122 runs were used to develop the best set of factor levels and evaluate main effects and two-way interactions among the 12 main components of the extender. A coordinate-exchange algorithm was used to construct the I-optimal design.

We used Design Expert software (version 13, 2021) to build runs and do analyses of variance for total and progressive motilities. The partial least squares (PLS) regression, cluster analysis, and Monte Carlo simulation were done by using the Microsoft Excel XLSTAT program (Version 2019.2.2.59614). Multiple factor analysis (MFA) and PLS path modeling (PLS-PM) were performed with R (version 4.2.2, R Core Team, 2022), utilizing the “FactoMiner” [17] and “plspm” [18] packages, respectively.

The Statistical Analysis System (SAS) version 9.4 (SAS Institute, Cary, NC, USA) was used to find outliers in a set of data that included all the observations from four experiments (replications). To figure out which observations were outliers, residuals were calculated for each one, and box plots were used to get rid of the 88 observations with the highest residuals. First, a random number function in Excel was used to mix up 400 data points. Then, the data were split into two groups: the calibration data subset (75%, 300 records) and the validation data subset (25%, 100 records). Data subsets have been used to determine the efficiency of PLS models [12]. The coefficient of determination (R^2^), root mean square error (RMSE), mean absolute percentage error (MAPE), and relative percent difference (RPD) were used to measure the effectiveness of models (Equations (1)–(4)) and were defined as:(1)$$R^{2}=1-\frac{\sum_{i}^{n}{\left(M_{i}-P_{i}\right)}^{2}}{\sum_{i}^{n}{\left(A_{i}-P_{i}\right)}^{2}},$$
(2)$$\mathrm{RMSE}=\sqrt{\frac{1}{n}\sum_{i=1}^{n}{\left(P_{i}-M_{i}\right)}^{2}},$$
(3)$$\mathrm{MAPE}=\frac{100}{n}\overset{n}{\underset{i=1}{\sum }}\frac{1}{M_{i}}\left|M_{i}-P_{i}\right|,$$
(4)$$\mathrm{RPD}=\frac{\mathrm{SD}}{\mathrm{RMSE}},$$
where *P_i_, M_i_, A_i_*, and SD are the values of the predicted, measured, average, and standard deviation of total motility or progressive motility, and *n* is the number of measuring points.

For PLS regression, the data included two dependent (response) variables: total and progressive motilities, as well as 12 independent (explanatory) variables: water, tris, egg yolk, milk, fructose, trehalose, CLC, glutathione, melatonin, NGF, glycerol, and ethylene glycol. The analysis was done in steps, such as making a correlation matrix between the variables, testing the model’s quality based on the number of components, and estimating the model’s parameters so that prediction equations for both total and progressive mobilities could be made. Jackknife cross-validation was performed, and bulls were taken into account in the PLS. The usual loading plot (*w*/**c* map) of PLS regression is presented (Figure 1). Using the agglomeration method of the unweighted pair-group average (UPGMA) and the Euclidean distance, a cluster analysis was done on three PLS components (*t*_1_, *t*_2_, and *t*_3_) to evaluate variations and similarities among the bulls.

In order to estimate trends and the expected magnitude of differences in motility for different media components and their variation, a Monte Carlo simulation model was made using 12 independent variables with normal distributions and one result for either total motility or progressive motility (multiple linear models) from the PLS. The component concentrations of the extenders were determined by 10,000 simulations.

An MFA can be used to account for the variable distribution in different subspaces that they generate [17]. On the MFA, the analysis of the groups of variables on manifolds or complementary (global analysis) is balanced, and each group’s structure is respected. In fact, the MFA analysis can be used to show how the sets of variables in a common space relate to each other. Multiple-factor analysis takes the complex data of the correlation matrix, which is often hard to understand and reduces it to a smaller number of factors (dimensions) [19].

Fourteen different parameters were put together into new active continuous sets of variables called “basic compound”, “extracellular protectant”, “CPA”, “osmoregulator”, “antioxidant class”, and “motility” (cf. Figures 6 and 7). This was done to group parameters that work well together and represent a complementary or global analysis. One supplementary categorical group, “bulls”, was added to the groupings to help with the analysis’s interpretation. MFA analysis can be supplemented by the PLS-PM method, which uses simple regressions to find out more about latent variables. PLS-PM is a combination of two models: a measurement model (also called an “outer model”) and a structural model (also called an “inner model”). The measurement model shows how manifested variables relate to latent variables in blocks or tables. Each block is made up of manifest variables and represents a latent variable. Using linear regression, the structural model explores the connections among latent variables (Sanchez, 2013).

In the present study, the measurement model had five blocks: “basic compound”, “extracellular protectant”, “CPA and osmoregulator”, “antioxidant class”, and “motility”. Each block included a latent variable and the corresponding manifested variable. According to Sanchez’s (2013) method, correlations between variables defined the construct type (reflective or formative). Positive and significant correlations are required for reflective constructs, while they should be avoided for formative constructs. Examining the unidimensionality of these blocks using Dillon-Goldstein’s rho confirmed this criterion. A block is unidimensional if its rho value exceeds 0.70 [18]. The structural model specified the block-to-block linkages based on our prior review [3]. The other blocks don’t explain the exogenous latent variables, but they do explain the endogenous latent variables. To validate that the blocks were correct, the PLS-PM model was used to figure out the R^2^ coefficient of each exogenous block, which showed how much variation each block could explain. When the R^2^ coefficients are high, other latent factors better explain the block. The link between blocks was clarified via path coefficients. The robustness of the model was finally assessed using the goodness-of-fit test [18].

## 3. Results and Discussion

In this study, we used different multivariate statistical approaches to find the best combination of media components for bull semen cryopreservation media. Each tool looked at relationships from a different angle and had different strengths and challenges.

With I-optimal design, PLS, and Monte Carlo simulation, less than 500 different experimental datasets were used to find the best media composition. These methods were necessary for many experiments to find the best way to make the above compositions. These methods have been used in several fields, including agriculture, the food industry, remote sensing, physics, etc. [10,12,13,14]. In our previous study, Tu et al. [9] used artificial neural networks and Gaussian process regression, supervised learning models, to optimize the media components of bull sperm cryopreservation.

We analyzed how the MFA and PLS-PM methods show the direct, indirect, synergistic, and opposing effects. Our MFA suggested that some antioxidants may have more beneficial effects on motility. We also found that the immediate effect of some antioxidants on motility may have been negative.

A fit summary suggested a linear regression between motilities and components (Table 1). The models were both significant (*p* < 0.01) and reliable (R^2^ > 0.8). The main effects of milk, fructose, and trehalose, and the two-way interactions of water × melatonin, tris × melatonin, milk × fructose, milk × trehalose, and trehalose × glutathione on progressive motility were significant (Table 1). Analysis of variance showed only a two-way interaction between CLC and melatonin (Table 1). Higher-level interactions may be significant when the main effects and two-way interactions are insignificant. This means some components can neutralize the effects of other ingredients, and some compounds, especially those with the same role, should be screened for future experiments. Similar main and interaction effects were found by Ramesh Pathy et al. [4], who reported that the effects of egg yolk, vitamin C, and glucose on the motility of human sperm were significant, and by Dorado et al. [20], who observed that a higher concentration of egg yolk made cold storage of dog sperm more effective for preservation. We are unaware of other studies that have examined more than two factors simultaneously.

Various quantiles of total motility measured for extenders obtained from the I-optimal design are presented in Table 2. Extenders 1 and 7 had the minimum and maximum total motility and progressive motility, respectively. The critical message of Table 2 is that we expect to observe higher total and progressive motility with more fructose, trehalose, and glycerol and less NGF and ethylene glycol.

Relationships between motilities as dependent variables and 12 components as independent variables, as well as coefficients of the models for total motility and progressive motility projection, are shown in Table 3. These equations were then used for Monte Carlo simulations. Water, egg yolk, glutathione, melatonin, NGF, and ethylene glycol have negative coefficients. This means that using these compounds in large quantities has harmed motilities. Considering these parts’ direct and indirect effects simultaneously is essential [21]. The PLS models had high precision (R^2^ ranged from 0.64 to 0.70) and accuracy (RMSE ranged from 7.35 to 7.76) (Table 3).

The independent variables closest to the dependent variables (total and progressive motilities) are assigned as positive effect components. In contrast, those far from the dependent variables are given as negative effect components (Figure 1). When PLS loadings (component 1 vs. component 2) were analyzed, it was found that the variables trehalose, fructose, CLC, glycerol, milk, and tris were all positively linked to total and progressive motilities.

Figure 2 shows three PLS components (*t*_1_, *t*_2_, and *t*_3_) used for cluster analysis, showing four main groups of bulls. The first group consisted of 28 bulls; the second, 10; the third, three; and the fourth group, only one bull. This shows that the viability after thawing varies from one sample to the next, and according to the most significant group, only 67% of bulls are nearly the same. These differences between these cohorts of bulls can be explained by the inherent inter-individual genetic variations and ejaculates within each bull. To understand whether the ejaculates within each bull are significantly different, samples at other time points must be taken from the same bulls. Salinas et al. [22] found that the response of swamp buffalo sperm to freeze-thaw procedures was not always the same among bulls. In contrast, however, Cardoso et al. [23] found that, in dogs, there was no significant within-male or among-male difference in post-thaw semen motility. This factor is vital because Argiris et al. [11] found that the quality of frozen semen production was the highest determinant of the superiority of any individual bull.

The associated values of R^2^ suggest that the PLS models could predict 74% and 67% of total and progressive motility variations, respectively (Figure 3). These were excellent quantitative models based on RPD. RPD values of about 2 are ideal for evaluating models [12].

The typical deviation is the standard deviation of a random variable with a Gaussian distribution. In the natural sciences, normal distributions are often employed to describe random variables with actual values whose distributions are unknown. We expect a decrease in marginal productivity as levels of the limiting factor rise. In Figure 4, there is a degree of distortion (skewness) in the symmetrical bell curve for components that indicates whether higher or lower quantities are required. In other words, when some parts are more asymmetrical than others, less or more of those parts are needed. For example, ethylene glycol has a right (positive) skewness, which means a lower concentration is necessary. We observed that glycerol has a left (negative) skewness, which means we can use higher concentrations in future experiments. In addition, we can limit the range of components. For example, the optimal content for egg yolk is 1.8 to 2.0 mL.

Ethylene glycol and NGF had the most unfavorable impacts on total and progressive motility, respectively (Table 4), while glycerol and trehalose had the most favorable results. The cryoprotective effects of glycerol and trehalose on the motility parameters of bull sperm have been shown in different studies [24,25,26,27,28]. In contrast, Foote et al. [29] discovered that trehalose did not increase the fertility of bull sperm frozen in whole milk. As a mechanism, Xu et al. found that adding trehalose can change sperm amino acid synthesis and the glycerol metabolism pathway [30].

The components and concentrations of top performing extenders are shown in Table 5 and Table 6. The total motility of top extenders ranged from 71.3% to 73.3%, and progressive motility varied from 58.4% to 65.7%.

The MFA model was used to determine which parameters are more important for increasing motility. This evaluation is shown by the “bulls” supplementary variable. According to the MFA scatter plots, the first two dimensions explain 45.42% of the total variance (Figure 5). In the MFA contribution plot, the “motility”, “CPA”, and “osmoregulator” groups have a strong correlation with dimension 1, while the “antioxidant class”, “basic compound”, and “extracellular protectant” groups have a strong correlation with dimension 2. The supplementary variable, “bulls”, does not alter the MFA dimensions, but supports the explanation of the findings. In fact, in this figure, the “bulls” variable is immediately next to motility and shows that motility isn’t always the same among bulls and there is significant bull-to-bull variation. Reports show that bull breeds have a significant effect on sperm quality and that different breeds need different measures of sperm quality [15].

The current study proposed that CPA and osmoregulator parameters had a significant favorable effect on motility. This is supported by the fact that they have a higher correlation with dimension 1. The correlation between dimension two and “antioxidant class” was more robust, suggesting that some antioxidants may have helped increase motility. Considering the biplot at the single parameter level informs how the different variables are connected (Figure 6). The strongest correlation with dimension one was found between total motility, glycerol, progressive motility, trehalose, fructose, milk, and melatonin, in that order (Figure 6). However,, glutathione, melatonin, tris, NGF, CLC, and egg yolk, in that order, had the most vital positive relationship with dimension 2. Most parameters were linked to dimension 1; both dimensions had melatonin in the joint.

According to the MFA data, determining which factors have the most vital relationships with motility remains to be determined. With this analysis, we cannot decide on and measure the relationship between the factors and the post-thaw viability of the sperm at the levels of a single variable and a global construct. Therefore, a PLS-PM model was used to study how well single and group variables fit together. It must be noted that the primary objective of using different methods to judge the quality of post-thawed sperm is to improve the synergy between parameters so that better predictions can be made. The overall goodness-of-fit for the PLS-PM was 0.427. In other words, about 57% of the total difference can’t be explained. This could be because the bulls being considered are different or because low or high amounts of some components are insufficient for motility.

The findings obtained in the MFA are supported by Figure 7. The parameter with the greatest loading value among antioxidant markers was glutathione. Glycerol had the highest loading of all the CPA and osmoregulator parameters. The largest loading among essential compounds was associated with fructose. The loadings of milk as an extracellular protectant, water as a primary compound, and ethylene glycol as CPA were negative. Although egg yolk had a favorable loading, it showed a low loading value. The correlation coefficients between the manifest and corresponding latent variables were lower than 0.5 for the melatonin variable in the antioxidant block, the tris variable in the basic compounds block, and the egg yolk variable in the extracellular protectant block (Figure 7). Öztürk [27] pointed out that freezing extenders containing modest concentrations of cryoprotectant combined with trehalose reduced the sensitivity of ram spermatozoa to cryopreservation and osmotic damage.

Except for the extracellular protectant block, the R^2^ value for all endogenous blocks was greater than 0.2 (Figure 8). The R^2^ coefficients for the latent endogenous variables were ordered as follows: motility > CPA and osmoregulator > antioxidant class > extracellular protectant (Figure 8). The “motility” block was positively linked to the “CPA and osmoregulator” block (0.64) and the “basic compound” block (0.16), but it was negatively related to the “antioxidant class” block (−0.19) and the “extracellular protectant” block (−0.13). The “CPA and osmoregulator” block was positively correlated with all other blocks. The “antioxidant class” block had a positive correlation with the “CPA and osmoregulator” block (0.42) and the “extracellular protectant” block (0.16). There was a negative relationship between the “basic compound” block and the “extracellular protectant” block (−0.31). We expected to see positive direct effects of antioxidants on motility but did not. Thus, removing antioxidants with lower loadings like melatonin and NGF may be advantageous, with the possibility of substituting more powerful antioxidants like proline [31] and humanin [1].

In this study, we aimed to develop the best-performing extender while exploring the interactions between various ingredients. Extender formulations play a significant role in the standard cryopreservation protocol, and each ingredient has a specific function such as cryoprotectant, antioxidant, energy metabolism function, and membrane function during freeze-thaw [3]. We observed multiple interactions between each individual component, and this is not possible to discover with conventional experimentation such as comparing two components at a time [9]. With our multivariate statistical approaches, we observed high within- and among-bull post-thaw total motility and progressive motility from sample to sample.

This is not surprising given that each ejaculate was from a different bull and given that each ejaculate has different biochemical signatures and physiological characteristics [32]. The fact that each individual bull showed a different reaction to a standard freezing protocol has been previously reported in the boars [33], horses [34] sheep [35], and dogs [36]. This phenomenon is independent of breed, genetic background, or diet [37]. However, differences in genomes, proteome, and epigenomes have been identified as markers of variation. Our multivariate statistical approach can be used as a robust tool to predict widely varying post-thaw fertility within and among bulls..

Our multivariate approach predicted a small variation in the post-thaw motility even though the optimal ranges for some of the components, such as NGF, were large. This may indicate that the interaction effects between NGF and other compounds are larger than other mutual compounds. This is aligned with a study in which NGF was reported as a detrimental factor for semen cryopreservation [38]. Therefore, many other factors, such as types and concentrations of ingredients as well as processing, can have an impact on NGF competency. It must be noted that semen samples are extremely heterogeneous, with sampling day, breed difference, and seasonal variation producing samples with large variability [39].

Additionally, in one of our previous studies, we showed that antioxidants and other ingredients of the extender can only provide better protection to semen with poor pre-freeze quality [40]. These pre-freeze quality indicators include a wide range of parameters such as motility, viability, membrane functionality, and, importantly, the antioxidant level of seminal plasma. It has been shown that semen with sufficient antioxidant levels is better able to resist freeze-thaw stress and therefore might not need additional antioxidants or other protectants [41].

We selected total motility and progressive motility parameters to compare different extenders because they are the main quality control parameters for selecting semen with the highest fertility potential [42]. There is a positive significant correlation between motility and fertility. In our study, post-thaw total motility and progressive motility were quite comparable to our standard control extender that is regularly applied for artificial insemination. The post-thaw motility obtained by this approach was comparable to commercial and industry standards; however, only a field fertility trial will provide complete evidence of the true fertility of each sample as a function of the preservation medium.

## 4. Conclusions

Even though there was a positive relationship between total motility and some media components, some of these components had a negative direct effect on motility. To design protocols for future experiments, the types and amounts of antioxidants and extracellular protectants must be changed and improved. In addition, milk, ethylene glycol, and water, in that order, had direct and indirect negative effects on total motility, which should be assessed to determine whether the toxic effects are due to type, amount, or both. The correlation between motility and milk was positive, but the direct effect of extracellular protectants on motility was negative. This negative effect can be explained by the fact that milk has a high negative loading and egg yolk has a very low positive loading. The direct effect of basic compounds on motility was positive. This can be attributed to the positive loadings of fructose and tris, but not to the negative loading of water. Glycerol and trehalose showed positive correlation coefficients with motility and had the highest positive path coefficient as an “CPA and osmoregulator” block with them. In contrast, ethylene glycol had a significant negative effect on motility.

## Figures and Tables

**Figure 1 animals-13-01077-f001:**
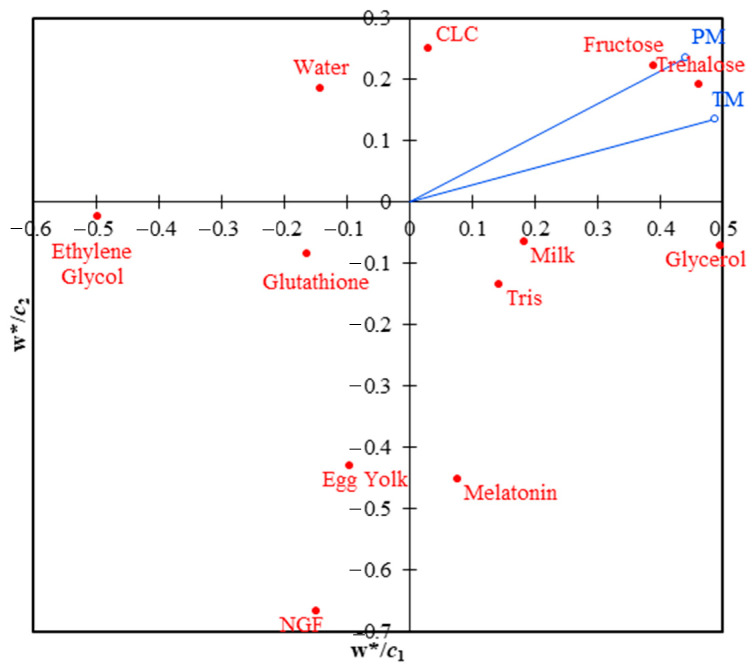
PLS variable loading plot for the two first components (*w**/*c*_1_ and *w**/*c*_2_), explaining how the Y variables (TM: total motility and PM: progressive motility) correlate to the X variables and the correlation structure.

**Figure 2 animals-13-01077-f002:**
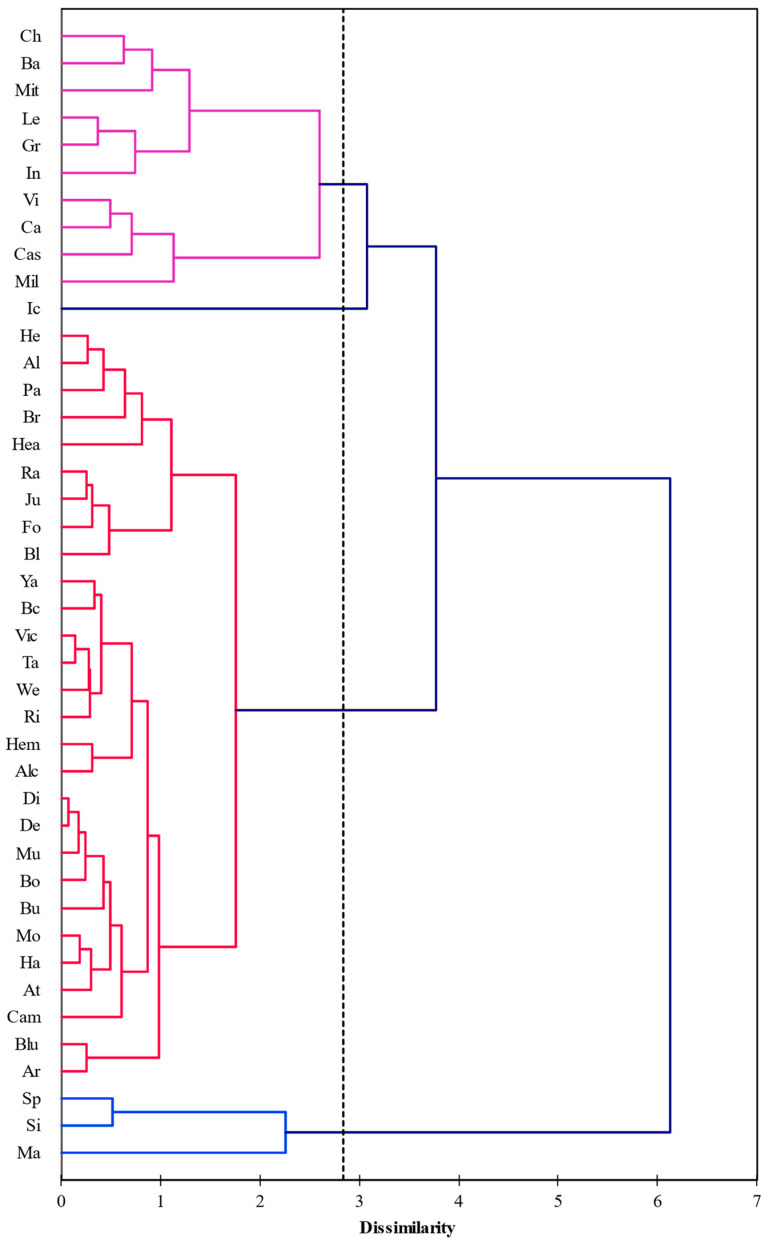
Dendrogram generated by the cluster analysis on three PLS components (*t*1, *t*2, and *t*3) shows the relationships among the studied bulls. Different colored lines show four main groups of bulls.

**Figure 3 animals-13-01077-f003:**
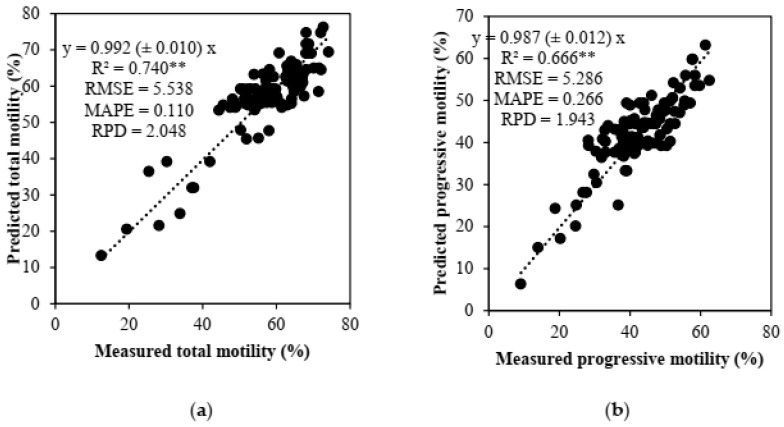
Relationship between predicted values of total (**a**) and progressive (**b**) motility and measured data (*n* = 100). Predicted values are generated by the PLS regression models. The model’s performance for PLS is shown by R^2^ (coefficient of determination), RMSE (root mean squared error), MAPE (mean absolute percentage error), and RPD (relative percent difference). The numbers in parentheses are the standard errors. ** *p* ≤ 0.01.

**Figure 4 animals-13-01077-f004:**
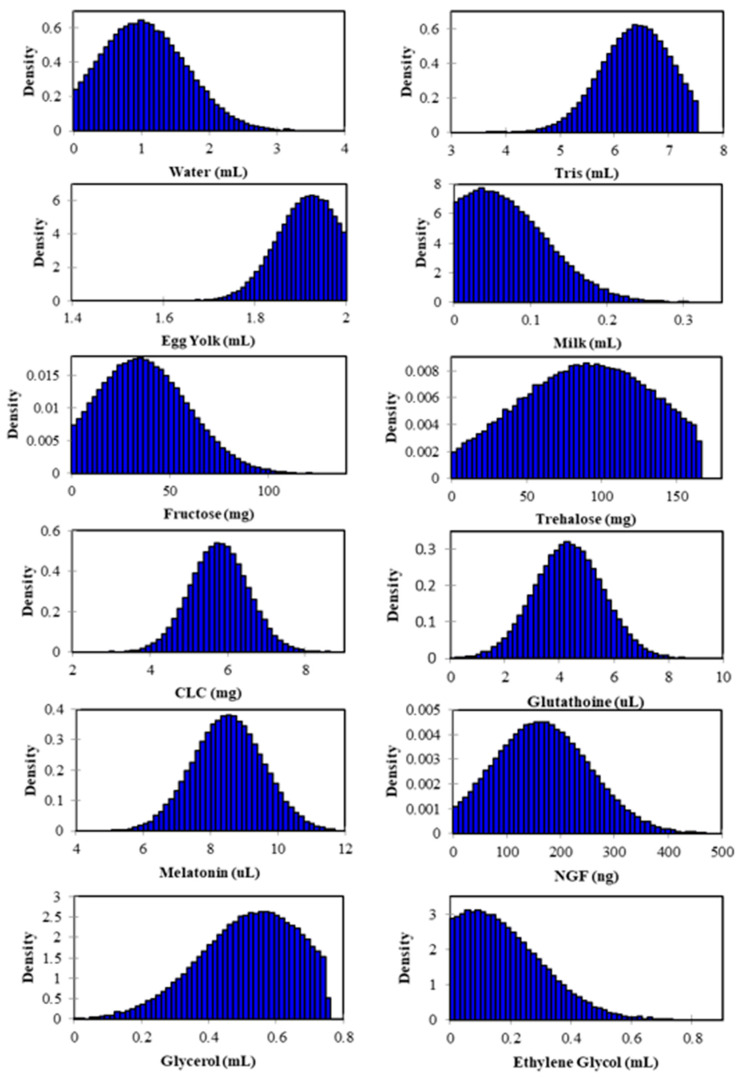
Frequency histogram and normal distribution density function obtained from the Monte Carlo simulation (*n* = 10,000) for the concentrations of components in the extender media in bull semen cryopreservation.

**Figure 5 animals-13-01077-f005:**
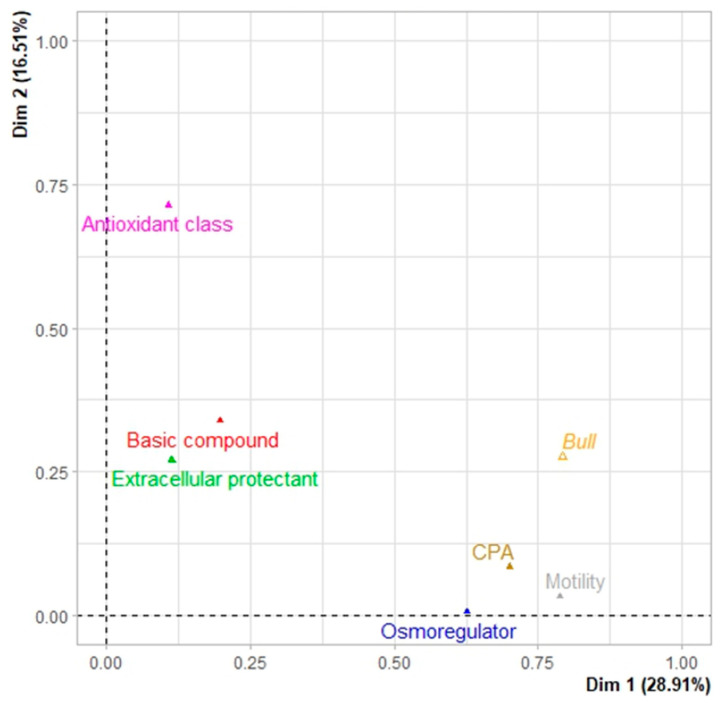
A contribution biplot of composite parameters on dimensions 1 and 2 after analysis of active groups (basic compound, extracellular protectant, CPA, osmoregulator, antioxidant class, and motility) and the supplementary variable (bulls).

**Figure 6 animals-13-01077-f006:**
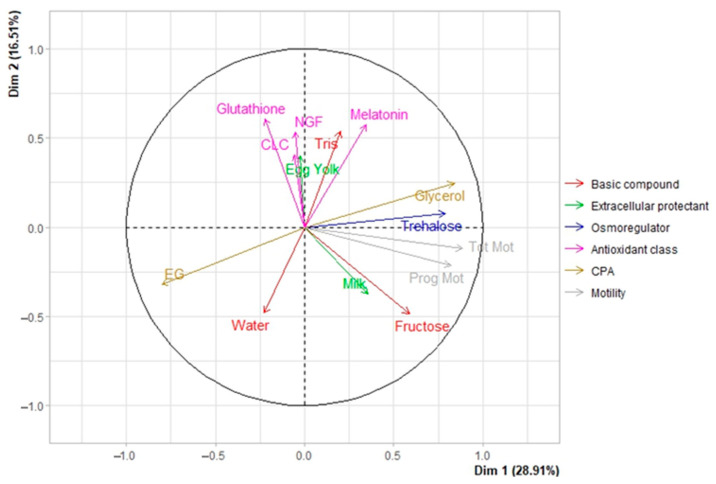
Biplot of a basic compound, extracellular protectant, CPA and osmoregulator, antioxidant class, and motility parameters on dimensions 1 and 2.

**Figure 7 animals-13-01077-f007:**
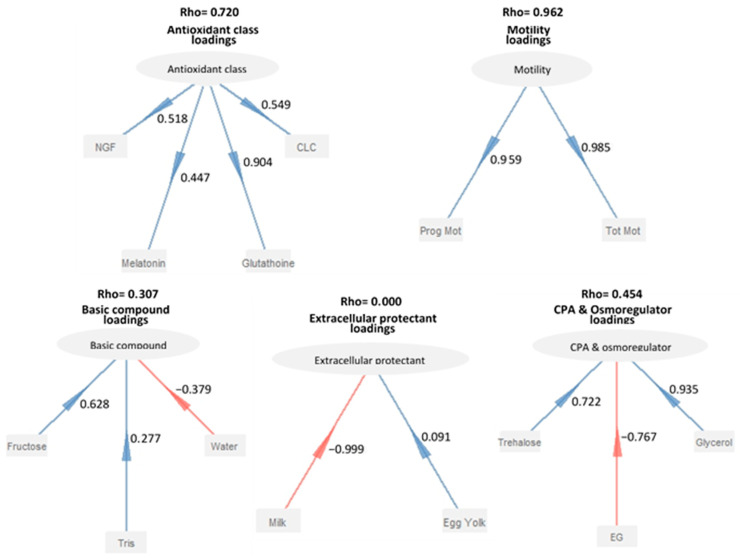
Loadings between basic compounds, extracellular protectants, CPA and osmoregulators, antioxidant classes, and motility and their relative parameters in PLS models. The values above each block are the Dillon-Goldstein rho (DG rho).

**Figure 8 animals-13-01077-f008:**
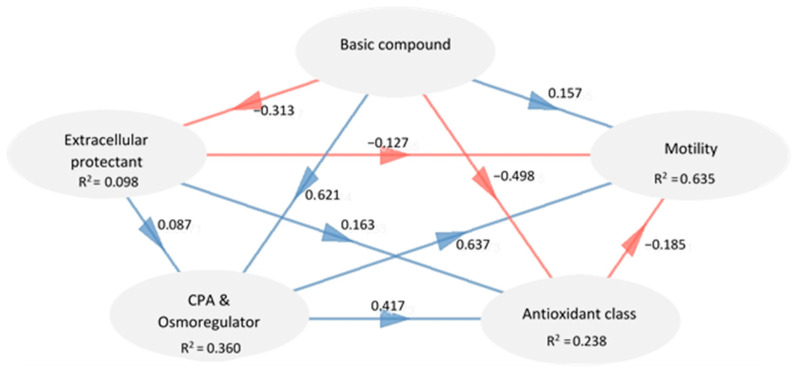
The structural model represents the network of interactions between blocks. Each arrow represents its direct coefficient. Blue and red arrows represent positive and negative coefficients, respectively. Endogenous blocks are also presented with their R^2^ coefficient.

**Table 1 animals-13-01077-t001:** Analysis of variance (Type III-Partial) of the main effects and some significant two-way interactions in terms of total motility and progressive motility.

		Total Motility			Progressive Motility	
**Source of Variation**	**Df**	**Mean Square**	***p*-Value**		**Mean Square**	***p*-Value**
Water	1	38.53	0.1103		17.49	0.2653
Tris	1	46.13	0.0814		2.48	0.6720
Egg Yolk	1	33.73	0.1344		3.43	0.6188
Milk	1	5.66	0.5352		99.29	0.0110
Fructose	1	1.61	0.7408		83.29	0.0190
Trehalose	1	1.21	0.7736		107.51	0.0084
CLC	1	36.98	0.1175		3.13	0.6346
Glutathione	1	37.52	0.1149		11.31	0.3687
Melatonin	1	7.83	0.4664		31.27	0.1394
NGF	1	52.38	0.0640		3.60	0.6104
Glycerol	1	41.57	0.0976		0.2479	0.8934
Ethylene Glycol	1	32.84	0.1395		0.1680	0.9122
Water × Melatonin	1	13.09	0.3471		59.69	0.0443
Tris × Melatonin	1	13.10	0.3470		59.45	0.0448
Milk × Fructose	1	13.99	0.3313		107.70	0.0084
Milk × Trehalose	1	12.17	0.3646		106.24	0.0088
Trehalose × Glutathoine	1	12.21	0.3637		62.24	0.0403
CLC × Melatonin	1	79.22	0.0241		43.13	0.0846
Residual	43	14.49			13.59	
**Fit Summary**	**Sequential *p*-value**	**R^2^**		**Sequential *p*-value**	**R^2^**	
Linear	0.0001	0.8709	**Suggested**	0.0001	0.8344	**Suggested**
Quadratic	0.6851	0.9227		0.0283	0.9024	**Suggested**
Cubic			**Aliased**			**Aliased**

**Table 2 animals-13-01077-t002:** Minimum, maximum, and percentiles of the treatments in terms of total motility obtained from the I-optimal design. In addition, for the same treatments, the values and standard deviation (SD) of progressive motility were presented.

**Components**	**Extender 1**	**Extender 2**	**Extender 3**	**Extender 4**	**Extender 5**	**Extender 6**	**Extender 7**
Water (mL)	1.68	2.67	1.05	0.40	0.28	0.27	1.60
Tris (mL)	5.42	4.75	6.43	7.09	6.99	7.01	5.88
Egg Yolk (mL)	1.79	1.88	1.88	1.88	2.00	1.87	2.00
Milk (mL)	0.00	0.00	0.00	0.00	0.17	0.17	0.00
Fructose (mg)	0.10	0.14	58.85	44.14	43.64	43.64	59.77
Trehalose (mg)	0.07	0.08	101.10	79.64	81.92	81.92	143.29
CLC (mg)	5.82	4.70	5.97	5.18	5.19	5.19	5.15
Glutathione (uL)	5.00	4.00	5.00	3.00	3.10	3.10	3.00
Melatonin (uL)	8.00	8.00	10.00	8.00	8.40	8.40	8.30
NGF (ng)	203.00	121.00	245.19	213.10	211.66	211.66	0.00
Glycerol (mL)	0.19	0.00	0.60	0.60	0.60	0.60	0.60
Ethylene Glycol (mL)	0.85	0.60	0.00	0.00	0.06	0.06	0.00
**Traits**	**Minimum**	**5th percentile**	**1st quartile**	**Median**	**3rd quartile**	**95th percentile**	**Maximum**
Total motility	0.67	18.01	50.61	57.25	63.53	68.31	70.56
Total motility SD	0.19	14.20	4.95	4.93	6.05	2.42	3.41
Progressive Motility	0.00	12.47	39.67	44.41	34.08	51.93	52.06
Progres. Motility SD	0.00	11.19	4.28	1.35	9.77	2.70	6.19

**Table 3 animals-13-01077-t003:** The parameters generated by the PLS regression models and the goodness of fit statistics for total and progressive motility (*n* = 300).

Variable	Total Motility (%)	SD	LB (0.95)	UB (0.95)	Progressive Motility (%)	SD	LB (0.95)	UB (0.95)
Intercept	72.329	10.08	47.66	97.00	76.064	8.86	54.38	97.74
Water (mL)	−1.943	0.98	−4.34	0.45	−1.272	0.94	−3.58	1.04
Tris (mL)	2.366	0.89	0.18	4.55	1.800	0.79	−0.14	3.74
Egg Yolk (mL)	−26.686	6.11	−41.64	−11.73	−30.753	4.84	−42.60	−18.91
Milk (mL)	14.580	11.35	−13.19	42.35	8.994	13.29	−23.52	41.51
Fructose (mg)	0.114	0.02	0.06	0.17	0.099	0.02	0.04	0.16
Trehalose (mg)	0.071	0.01	0.05	0.09	0.061	0.01	0.04	0.08
CLC (mg)	2.500	1.48	−1.12	6.12	2.921	1.26	−0.16	6.01
Glutathione (uL)	−0.964	0.50	−2.18	0.25	−0.802	0.51	−2.05	0.45
Melatonin (uL)	−0.849	1.04	−3.39	1.69	−1.464	0.92	−3.72	0.80
NGF (ng)	−0.030	0.01	−0.05	−0.01	−0.035	0.01	−0.05	−0.02
Glycerol (mL)	21.724	3.24	13.80	29.64	16.090	3.04	8.65	23.53
Ethylene Glycol (mL)	−23.890	2.57	−30.19	−17.59	−19.353	2.14	−24.60	−14.11
Goodness of fit statistics								
R²	0.697				0.639			
RMSE	7.762				7.349			

SD: Standard deviation; LB: Lower bound; UB: Upper bound.

**Table 4 animals-13-01077-t004:** Correlations between components and total motility (TM), progressive motility (PM), and the sensitivity analyses using the Monte Carlo simulation.

Distributions	Correlation (TM)	Contribution (%, TM)	Correlation (PM)	Contribution (%, PM)
Water	−0.154	2.33	−0.104	1.08
Tris	0.208	4.26	0.171	2.92
Egg Yolk	−0.223	4.86	−0.283	8.03
Milk	0.103	1.03	0.076	0.57
Fructose	0.347	11.82	0.305	9.34
Trehalose	0.403	15.90	0.379	14.41
CLC	0.258	6.53	0.302	9.18
Glutathione	−0.193	3.65	−0.151	2.28
Melatonin	−0.125	1.52	−0.236	5.61
NGF	−0.347	11.80	−0.445	19.86
Glycerol	0.422	17.43	0.341	11.70
Ethylene Glycol	−0.439	18.88	−0.387	15.02

**Table 5 animals-13-01077-t005:** The components and concentrations of the ten top extenders for total motility determined by the Monte Carlo simulation.

Extender	Water (mL)	NGF (ng)	Glycerol (mL)	Ethylene Glycol (mL)	Tris (mL)	Egg Yolk (mL)	Milk (mL)	Fructose (mg)	Trehalose (mg)	CLC (mg)	Glutathione (uL)	Melatonin (uL)	Total Motility (%)
1	1.07	110.79	0.50	0.07	6.68	1.83	0.08	93.60	138.85	6.56	3.73	7.16	71.31
2	0.00	83.41	0.50	0.30	7.00	1.77	0.12	84.52	161.94	6.50	2.72	9.21	71.49
3	0.41	2.07	0.71	0.02	6.85	1.84	0.18	62.29	93.72	6.24	4.56	11.16	71.73
4	0.13	141.23	0.65	0.00	7.47	1.97	0.17	65.15	157.46	6.92	5.27	9.67	72.04
5	0.27	74.50	0.62	0.02	6.96	1.85	0.14	43.40	151.37	6.36	3.97	8.90	72.12
6	1.47	119.73	0.59	0.13	7.35	1.84	0.16	110.62	93.78	6.76	3.39	8.50	72.15
7	0.42	28.06	0.63	0.03	6.65	1.94	0.13	72.37	125.71	6.41	4.62	6.94	72.26
8	2.29	165.91	0.75	0.15	7.35	1.80	0.11	80.32	144.82	6.33	4.77	5.93	72.28
9	0.63	60.03	0.75	0.20	6.99	1.79	0.18	47.68	140.87	5.91	3.09	7.04	73.50
10	1.43	46.67	0.66	0.06	6.74	1.98	0.24	62.07	161.80	6.12	2.36	6.59	73.75

**Table 6 animals-13-01077-t006:** The components and concentrations of the ten top extenders for progressive motility driven by the Monte Carlo simulation.

Extender	Water (mL)	NGF (ng)	Glycerol (mL)	Ethylene Glycol (mL)	Tris (mL)	Egg Yolk (mL)	Milk (mL)	Fructose (mg)	Trehalose (mg)	CLC (mg)	Glutathione (uL)	Melatonin (uL)	Progressive Motility (%)
1	0.54	145.39	0.48	0.02	7.27	1.77	0.05	47.52	149.86	6.70	1.91	6.88	58.44
2	0.51	0.68	0.59	0.20	7.43	1.92	0.10	92.59	155.46	7.04	3.86	9.45	58.47
3	1.02	12.90	0.71	0.16	6.48	1.80	0.10	65.43	157.37	5.71	3.48	7.74	58.52
4	1.58	70.82	0.68	0.00	6.85	1.81	0.13	57.44	134.05	5.91	4.23	6.53	58.53
5	0.58	63.18	0.59	0.17	6.57	1.90	0.11	102.25	128.16	7.35	3.23	7.94	58.81
6	1.28	69.35	0.73	0.06	6.56	1.80	0.15	45.32	152.27	6.40	2.50	8.20	58.97
7	0.53	51.85	0.68	0.03	6.24	1.79	0.09	33.22	146.30	6.98	3.82	7.64	59.25
8	1.29	26.00	0.60	0.05	7.28	1.76	0.11	71.17	105.39	6.59	2.95	8.43	60.28
9	1.16	40.74	0.74	0.04	6.99	1.84	0.01	49.45	144.55	7.87	4.03	6.52	64.27
10	0.93	58.89	0.71	0.04	6.21	1.64	0.12	76.36	160.67	5.81	3.58	7.62	65.67

## Data Availability

Data are included as Appendix A.

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
