# Peer review of "Optimizing Bull Semen Cryopreservation Media Using Multivariate Statistics Approaches"

_animals, 2023, doi:10.3390/ani13061077_

Round 1

Reviewer 1 Report

This text describes the results and discussion of a study that used multivariate statistical approaches to determine the best combination of media components for bull semen cryopreservation media. The study found that some antioxidants had a negative effect on motility, while fructose, trehalose, and glycerol improved motility, and some components could neutralize the effects of other ingredients. Monte Carlo simulations were used to analyze the relationships between the components and the motilities, with some compounds found to negatively affect motility. The results showed that the viability of bull semen after thawing varied from one sample to the next, possibly due to inter-individual genetic variations. The text includes several tables and figures to present the results.

The text could be improved by:

1.     Improving the clarity and flow of the presentation. The information is presented in a dense and disorganized manner, making it difficult for the reader to follow the study's results and conclusions.

2.     Improving the text's readability.

3.     Highlighting the most important results and conclusions. The text contains much information, but it is unclear which results and conclusions are the most important to the study's objectives.

4.  Avoid repetition. Some information, such as the results of the Monte Carlo simulations, is repeated in different sections of the text.

5.     Improving the visual presentation. The tables and figures could be better labeled and positioned for clarity and ease of interpretation.

I also attach a file with writing suggestions in the results and discussion section

Reviewer 2 Report

The study examines a difficult and important topic in spermatology: finding the optimal semen extender composition for cryopreservation of bull sperm. The approach of examining very different extender compositions with the help of in-depth statistical methods makes sense, especially to clarify relationships between the individual components. However, in the material and methods for the experimental work are some information lacking (see below in detail), a more detailed descriptions of the experimental design is necessary. Additionally, after the long and detailed description of the statistical results the summarizing conclusions from the various statistical tests are too brief and, above all, should be discussed in more detail with regard to the physiological significance of the individual components and also with the existing knowledge from the literature.

Special comments:

Experimental design:

-        Why was only the total motility and progressive motility put into the evaluation of the semen quality after thawing? Of course, these parameters are related to fertility, but regarding some the various components, the proportion of acrosome-reacted sperm would also be of particular interest. This should at least be discussed.

-        In the material and methods you give for water and other components a range in mL. How was the basic volume of the extender, to which this volume of extra water was added? As you give (line 109 and 110) the basic extender components in mM, I would suggest, that there is surely already water in your standard control extender. Please specify.

-        Line 111: What means 3-8mL tris? What was the concentration of this tris? Please specify.

-        Line 110 to 114: You give lower and upper limits for the ranges of the added components - In which/how many individual steps were these ranges subdivided?

-        Line 118: When did you add glycerol and ethylene glycol to the semen? Before cooling or at a defined time during cooling? That would have different effects.

-        As stated above, please specify the experimental design in more detail: How many ejaculates were collected per bull? Did you do split-samples for the same ejaculate with different extender compositions? Since the bulls and individual ejaculates can vary greatly in terms of motility (as it is also described in the manuscript), this has a major impact on the comparability of the individual motility results. Maybe a table or figure for the experimental design would help understanding.

-        Analysis of the motility: How many videos (fields of view) were collected per individual sample and used for the motility analysis?

-        How long was the time between thawing and motility analysis? How long and at which temperature did you incubate? How many straws were thawed per individual sample and were these pooled or analyzed separately?

-        Why did you choose 33°C for the motility measurement instead of body temperature (38°C)?

Statistical analysis:

-        Why did you use four different statistical analysis programs?

-        Line 144: What means replicates here? Does it mean the same ejaculate from the same individual bull diluted in the same extender composition?

-        Line 144: Why did you choose to delete this 88 outliers – that seems a high number and maybe especially this outliers are no real outliers and would have been a useful addition to the result.

Results and discussion:

-        Table 1: Since total motility and progressive motility usually correlate, it is a bit strange that you found always only for one (total motility) or the other (progressive motility) a significant effect. I know, that is the results, but please discuss this.

-        Line 265 to line 271: this is a repetition from line 246 to 253. Please delete one of the two paragraphs.

-        Table 3: Why is the intercept of progressive motility higher than the intercept of total motility (as usually the total motility has the higher values)?

-        Table 5: There are sometimes very large differences in the optimal addition of individual extender component, for example NGF varies from 2.07ng to 141.23ng. Maybe this has something to do with the rest of the composition of this extender (agonistic/antagonistic effects). Please discuss this.

As stated above the physiological significance of the results (optimal ranges of components, links between components etc.) should be discussed in more detail. For your in-depth statistical analysis this should also be done in more depth. Additionally, this individual and combined effects of the different components should be discussed in more detail with the present literature.

Reviewer 3 Report

Utilizing artificial insemination of cryopreserved bull semen is greatly critical for the cattle industry. Therefore the quality of the frozen-thawed bull semen is of great importance. The cryopreservation of bull semen has been extensively studied in previous researches by investigation a few parameters. This manuscript employed multivariate statistical methods on a large experimental dataset over dozen compounds to identify the key compounds and their key concentrations. The experimental design, statistics, data interpretation and discussion all sounds good, and the language expression of the manuscript is appropriate.
